# The Dual-Mode Transition of Myofibroblasts Derived from Hepatic Stellate Cells in Liver Fibrosis

**DOI:** 10.3390/ijms242015460

**Published:** 2023-10-23

**Authors:** Mengchao Yan, Ye Xie, Jia Yao, Xun Li

**Affiliations:** 1Department of General Surgery, The First Hospital of Lanzhou University, Lanzhou 730000, China; 2The School of Medical, Lanzhou University, Lanzhou 730000, China; 3Key Laboratory of Biotherapy and Regenerative Medicine of Gansu Province, Lanzhou 730000, China

**Keywords:** liver fibrosis, hepatic stellate cells, transforming growth factor β, activation model

## Abstract

Hepatic stellate cells (HSCs) are the key promoters of liver fibrosis. In response to liver-fibrosis-inducing factors, HSCs express alpha smooth muscle actin (α-SMA) and obtain myofibroblast phenotype. Collagen secretion and high expression of α-SMA with related high cell tension and migration limitation are the main characteristics of myofibroblasts. How these two characteristics define the role of myofibroblasts in the initiation and progression of liver fibrosis is worth exploring. From this perspective, we explored the correlation between α-SMA expression and collagen secretion in myofibroblasts and the characteristics of collagen deposition in liver fibrosis. Based on a reasonable hypothesis and experimental verification, we believe that the myofibroblast with the α-SMA^high^collagen^high^ model do not effectively explain the initial stage and progression characteristics of liver fibrosis. Therefore, we propose a myofibroblast dual-mode transition model in fibrotic liver (DMTM model). In the DMTM model, myofibroblasts have dual modes. Myofibroblasts obtain enhanced α-SMA expression, accompanied by collagen expression inhibition in the high-concentration region of TGF-β. At the edge of the TGF-β positive region, myofibroblasts convert to a high-migration and high-collagen secretion phenotype. This model reasonably explains collagen deposition and expansion in the initial stage of liver fibrosis.

## 1. Introduction

Cirrhosis is a chronic liver injury induced by various factors, including metabolic disorders, viruses, and alcohol. The main characteristics of liver fibrosis are abnormal collagen deposition and structural destruction caused by multiple diseases, which further leads to portal hypertension and liver failure. Hepatic stellate cells (HSCs) are considered to be the main promoters of collagen deposition during fibrosis. Under the stimulation of inflammation and fibrogenic cytokines, HSCs express alpha smooth muscle actin (α- SMA) and transition from a quiescent state to myofibroblast phenotype [1,2,3].

Myofibroblasts are almost absent in normal tissues and are generated by resident fibroblasts, bone-marrow-derived progenitors, and possibly epithelial or endothelial cells in response to injury [4,5,6,7]. The activation of HSCs has been proven to be the main source of collagen-producing myofibroblasts in the liver [8,9]. Under physiological conditions, quiescent HSCs (qHSCs) exist in the Disse space, and their physiological functions include fat storage, the metabolism of vitamin A, and collagen secretion to maintain the stability of matrix structure [1,10]. Intercellular communication based on soluble factors provides stable communication and steady-state maintenance between qHSCs and adjacent cells, such as hepatocytes, cholangiocytes, Kupffer cells, liver sinusoidal endothelial cells (LSECs), and nerve cells [11]. The damage of epithelial cells in the liver and its induced inflammatory response are the key factors that induce the activation of HSCs. The injury of hepatocytes and cholangiocytes induces the aseptic inflammatory environment and promotes the infiltration of inflammatory cells, including macrophages, lymphocytes, and neutrophils [12]. Inflammatory-cell-derived TGF-β, PDGF, CTGF, IL-1 β, IL-6, IL-4, IL-13, TNF, and ROS are important activators of HSCs [3,11,13]. In addition, fibrosis-related matrix stiffness and metabolic disorders can further promote the myofibroblast phenotype of HSCs, forming a positive feedback loop [14,15,16,17].

α- SMA is the main marker of qHSCs activation and transformation into myofibroblasts, which promotes the contraction of myofibroblasts and the formation of traction. In the process of tissue repair, myofibroblasts are physically connected to ECM through integrins, which are associated with highly developed focal adhesion. “Supermature” focal adhesions and α-SMA expression promote the strong cell tension of myofibroblasts and promote wound closure [18,19]. Interestingly, α-SMA inhibits the migration of myofibroblast [20,21,22]. The enhanced adhesion and tension, combined with the reduced migration, promote the residence of myofibroblasts in the activated region. Due to the key role of aHSCs in the development of fibrosis, the collaborative mode of enhanced adhesion-tension and collagen secretion in aHSCs is closely related to the distribution of collagen in fibrotic liver. If aHSCs have strong adhesion-contraction force and enhanced collagen secretion at the same time, they will reside in the activated region and produce collagen, which will lead to the patchy distribution of collagen in a specific region. However, the distribution of collagen in a network, rather than in patches, is the main feature of liver fibrosis, which means that another accurate model needs to be proposed. Therefore, we discussed the combined effect of α-SMA expression and collagen secretion in HSC-derived myofibroblasts in the presence of an activating factor, and the results showed a negative correlation between α-SMA and COL1A1 expression in the presence of a TGF-β gradient. Based on the initiation and progression characteristics of liver fibrosis and the phenotypic evolution of myofibroblasts, we propose a dual-mode transition model of myofibroblasts in fibrotic liver (DMTM model). The DMTM model accurately describes the relationship between the characteristics of HSC-derived myofibroblasts and the development of liver fibrosis.

## 2. Results and Discussion

### 2.1. The Initiation and Progression Characteristics of Liver Fibrosis

In the early stage of liver fibrosis, the expression of TGF-β induced by ductular reaction in the portal vein region is the key factor in promoting the initiation and progress of fibrosis, and HSC-derived myofibroblasts are enriched in the portal vein region in response to TGF-β signal [23]. The combined enhancement of collagen secretion and α-SMA in myofibroblasts is a traditional model to explain the progression of fibrosis. We analyzed early liver fibrosis in rats (2 weeks of modeling), and the results showed that there was high expression of patchy TGF-β and mild fibrosis around the portal vein (Figure 1A,B). In addition, visible collagen deposition appeared in the liver lobules at the early stage of fibrosis, which was synchronous with portal vein fibrosis. Four weeks after modeling, liver fibrosis staining showed a uniform network without patchy collagen deposition in the portal vein region (Figure 1C). Therefore, the activated collagen deposition did not show a significant advantage in the portal vein region, and the combined enhancement model of collagen secretion and α-SMA of myofibroblasts could not effectively explain the progress of liver fibrosis.

### 2.2. Negative Correlation between α-SMA and Type I Collagen, Alpha 1 (COL1A1) Expression in Myofibroblasts in Single Cell Database

Based on the Homo sapiens fibrosis liver single-cell sequencing data set provided by P Ramachandran, we analyzed the correlation between α-SMA and COL1A1 expression in the cell population with myofibroblast characteristics [24]. With the help of the Omnibrowser database (https://omnibrowser.abiosciences.com/#/home) (15 May 2023), we obtained a total of 11 cell subpopulations, among which mesenchyme cells possessed the characteristics of myofibroblasts (Appendix A). The expression of myofibroblast markers in mesenchyme cells, including α-SMA, COL1A1, platelet-derived growth factor alpha polypeptide (PDGFA), and platelet-derived growth factor receptor beta polypeptide (PDGFRB), is higher than that of other subpopulations (Appendix A). In mesenchyme cells, the expression of α-SMA is negatively correlated with COL1A1 (Figure 2A). Next, we further validated this conclusion through models in vitro and in vivo.

### 2.3. Negative Correlation between α-SMA and COL1A1 Expression in HSC-Derived Myofibroblasts in the Presence of TGF-β1

Myofibroblasts are the main executors of migration and collagen secretion in the fibrotic liver. We further promoted the myofibroblast characteristics (α-SMA expression) of a rat HSC cell line HSC-T6 using 5% rat liver fibrosis serum (Figure 2B). After further treatment with gradient concentration TGF-β1, the expression of α-SMA mRNA in myofibroblasts was significantly increased, accompanied by a significant decrease in COL1A1 mRNA expression (Figure 2C). Western blot and immunofluorescence double staining further validated this conclusion at the protein level (Figure 2D, Appendix A and Appendix A). The negative correlation between α-SMA and COL1A1 expression in myofibroblasts is consistent with single-cell sequencing analysis. TGF- β/Smad signaling is the key pathway for activation of HSCs [25,26]. In the TGF-β gradient, the expression of p-Smad2/3 in HSC-derived myofibroblasts was enhanced (Figure 2D).

### 2.4. Differential Localization of α-SMA, COL1A1, and TGF-β in Rat Fibrotic Liver

We used carbon tetrachloride to construct the rat liver fibrosis model and evaluated the distribution of TGF-β, α-SMA, and COL1A1 in the tissue (Figure 2E). The staining results showed that the distribution of α-SMA positive myofibroblasts coincided with the TGF-β positive region, suggesting that the induction of myofibroblasts was associated with TGF-β. However, the significant positive regions of COL1A1 and TGF-β did not coincide. Therefore, it can be concluded that the activation of myofibroblasts in the high expression region of TGF-β does not lead to significant collagen deposition in situ, which occurs in the periphery of the TGF-β positive region. Fibrotic liver staining supports the conclusion of the cell model in vitro.

### 2.5. The Dual Mode Transition of Myofibroblasts in Fibrotic Liver (DMTM Model)

Based on the above data, we propose a DMTM model to explain the progression of liver fibrosis (Figure 3A). In the DMTM model, HSC-derived myofibroblasts contain model I (α-SMA^high^COL1A1^low^) and model II (α-SMA^low^COL1A1^high^). In the high-expression region of TGF-β, the dominant model I myofibroblasts reside and produce strong local tension through the high expression of α-SMA. The collagen secretion of myofibroblasts in model I is limited, so there is no patchy collagen deposition in the portal vein region at the initial stage of fibrosis. In the low-expression region of TGF-β, model II myofibroblasts were dominant and showed high collagen secretion and migration potential. The characteristics of the model II myofibroblasts surrounding the TGF-β expression region can provide a reasonable explanation for the early expansion of fibrosis and network-like collagen deposition. We further based the “α-SMA^high^ collagen^high^ model” on the tradition myofibroblast, and analyzed the initiation and development of liver fibrosis. In the region with high TGF-β expression, myofibroblasts simultaneously obtain collagen secretion and α-SMA-related contraction and adhesion, which may induce in situ colonization of myofibroblasts in the activated region and large amounts of collagen production. In this process, collagen will deposit in patchy form and continue to expand. Patchy collagen deposition is not consistent with the histological characteristics of liver fibrosis. Therefore, the analysis based on the activation mode of myofibroblasts showed the correctness of the DMTM model.

In addition, it is helpful to discuss the development of fibrosis in combination with the initiation of liver fibrosis and the response of myofibroblasts (Figure 3C). In steatohepatitis, “two hits” describes the two stages, namelysteatosis and lipid peroxidation, in the initial stage of inflammation [27]. The accumulation of reactive oxygen species and free radicals and the subsequent production of inflammatory cytokines induce the activation and expansion of myofibroblasts. Similarly, viral-hepatitis-related dysregulated T cell responses and mitochondrial stress provide the starting point for myofibroblast activation [28,29]. The activation of myofibroblasts combined with the TGF-β gradient initiated the deposition of collagen in the liver.

## 3. Materials and Methods

### 3.1. Cell Culture

Rat HSC cell line HSC-T6 was kindly provided by Procell Life & Technology, Wuhan, China. HSC-T6 was cultured in Dulbecco’s modified eagle’s medium (DMEM, GIBCO, Waltham, MA, USA) supplemented with 10% fetal bovine serum (FBS, Hyclone Laboratories, Logan, UT, USA) and 1% penicillin and streptomycin (Hyclone Laboratories, UT, USA). The cells were maintained at 37 °C and 5% CO_2_ in a humidified incubator during cell culture.

### 3.2. Real-Time Quantitative RT-PCR (qRT-PCR)

Total RNA was extracted from stomach tissues or cultured cells using Trizol solution (TAKARA, Japan). An amount of 2 μg of total RNA was used for the reverse transcription reaction with RevertAid First Strand cDNA Synthesis Kit (Thermo Fisher Scientific, Waltham, MA, USA). Real-time qPCR was carried out in 20 µL final volume using 1 ng/µL cDNA and forward and reverse primers (500 nM each) using TB Green Premix Ex Taq II (TAKARA, Asazu City, Japan) in CFX Connect PCR System (Bio-Rad, Hercules, CA, USA). The primer sequences are shown in Appendix A.

### 3.3. Western Blot

We washed the cells cultured in 6-well plates (Corning Life Sciences, Tewksbury, MA, USA) three times in PBS and lysed them on ice in a RIPA buffer, protease-inhibitor cocktail, and phosphatase inhibitor (APExBIO, Houston, TX, USA). Then they were mechanically broken down using a syringe. The suspension was centrifuged at 4 °C, and the supernatant was collected. The enhanced BCA protein assay kit (Beyotime Biotechnology, Shanghai, China) was used to determine the protein concentration of the supernatant. Proteins (50 μg) were separated on 10% SDS-polyacrylamide gels and transferred onto PVDF membranes (Millipore, Greenwich Township, NJ, USA). The PVDF membranes were blocked with 5% skim milk in TBS and then incubated with primary antibodies at 4 °C for 12 h. The anti-α-SMA antibody (1:2000) was obtained from Proteintech (14395-1-AP, Rosemont, PA, USA). The anti-COL1A1 antibody (1:1000) was obtained from PTMbio (PTM-6219, Hangzhou, China). The anti- Phospho-Smad2 (Ser465/467) + Smad3 (Ser423/425) antibody (1:1000) was obtained from Beyotime (AF5920, Shanghai, China). After three washings in TBS-T, the membranes were incubated with HRP-conjugated secondary antibodies (1:3000, PR30011, Proteintech, PA, USA) at 25 °C for 1 h and exposed to immobilon western chemilum HRP substrate (Millipore, USA). To confirm equal protein loading, we used an anti-β-actin antibody (1:2000, 20536-1-AP, Proteintech, PA, USA) to re-probe.

### 3.4. Elisa

The cell culture medium was centrifuged at 3000 rpm for 20 min. According to the manufacturer’s instructions, a total of 100 μL of standard, control, and supernatant was added to each well of a 96-well plate and incubated for 90 min at 37 °C. An amount of 100 μL biotinylated antibody was added to each well and incubated for 60 min at 37 °C. The plate was washed three times with the wash solution, and 100 μL 1× streptavidin-HRP solution was added into each well and incubated for 30 min at 37 °C. After washing steps, 90 μL TMB was added into each well and incubated for 10 min at 37 °C. Then, 50 μL stop solution was added. The absorbance was read at 450 nm after adding the stop solution.

### 3.5. Animal Treatment

Four-week-old male SD rats (n = 10) weighing 120–150 g were purchased from the Lanzhou Veterinary Research Institute of the Chinese Academy of Agricultural Sciences (Lanzhou, China) and carefully maintained under standard laboratory conditions, and were acclimated for 7 d before the experiments and allowed free access to food and water. The experimental rats were subjected to intraperitoneal injections of 2 mL CCl4/olive oil (1:1, *v*/*v*)/kg body weight 2 times per week to induce the liver fibrosis model. The state of the rats was regularly observed. All animals received humane care according to the criteria outlined in the Guide for the Care and Use of Laboratory Animals. All animal model schemes were approved by the Animal Care and Use Committee of the First Hospital of Lanzhou University (Ethics number: LDYYLL2020-280). After 2/4 weeks of modeling, rats were euthanized humanely under isoflurane inhalation, and livers were promptly dissected from the animal. Liver tissues were fixed in 4% neutral formaldehyde, embedded in paraffin, and sectioned. Sirius red staining was used for the analysis of histological structure and fibrotic area, respectively.

### 3.6. Sirius Red Staining

Sirius red staining is intended for use in the histological visualization of collagen I and III fibers. Sirius red staining was performed according to the manufacturer’s instructions (Abcam, ab150681, Cambridge, MA, USA), which included fixing fresh tissue, dehydrating, paraffin embedding, sectioning, deparaffinizing, hydrating, Sirius Red solution staining (60 min), rinsing with acetic acid solution and absolute alcohol, dehydrating, clearing slides, and mounting in synthetic resin. The images were acquired by Leica DM IL microscope (Bright filed) (Leica microsystems, Wetzlar, Germany).

### 3.7. Immunofluorescence

This experiment followed conventional methodology [30], which included fixing fresh tissue, dehydrating, paraffin embedding, sectioning, mounting on slides, and incubating with primary antibodies (anti-α-SMA, green, 1:250, 14395-1-AP; anti-TGF-β1, red, 1:250, 21898-1-AP; anti-COL1A1, pink, 1:200, 67288-1-Ig. These antibodies were bought from Proteintech) and corresponding secondary antibodies in sequence (Alexa Fluor 488-conjugated secondary antibodies, 1:200, GB25303; CY3-conjugated secondary antibodies, 1:200, GB21303; CY5-conjugated secondary antibodies, 1:200, GB27303; these antibodies were bought from Servicebio), and DAPI was used for the counterstaining of nuclei (Servicebio, Wuhan, China). In this process, the sections were incubated with primary antibodies at 4 °C overnight. The images were acquired by Leica DM IL microscope (Confocal) (Leica microsystems, Wetzlar, Germany).

### 3.8. Statistical Analysis

Data were analyzed using SPSS Statistics 25.0 (IBM, NYC, Armonk, NY, USA) and Prism 6.0 software (GraphPad Software, San Diego, CA, USA). Data are presented as the mean ± SD. Unpaired Student’s *t*-tests or Mann–Whitney U test were used to compare the means of the two groups. Specific statistical methods were selected according to the research purpose, normality, and homogeneity of variance of the sample. A *p*-value of <0.05 was considered statistically significant.

## 4. Conclusions

The portal vein region is the key starting point of liver fibrosis, which is related to the high expression of TGF-β in the portal vein region at the early stage of fibrosis. It is an interesting feature of early liver fibrosis that interlobular collagen deposition and the initiation of fibrosis in the portal vein region occur at the same time. If activated myofibroblasts have both high expression of α-SMA and enhanced collagen secretion, patchy collagen deposition centered on the portal vein region will be inevitable. Because of this contradiction, we analyzed the functional changes of HSC-derived myofibroblasts in response to TGF-β concentration gradient and proposed a DMTM model. The activation of myofibroblasts under a high concentration of TGF-β was mainly characterized by high α-SMA expression, while myofibroblasts in a low concentration of TGF-β were mainly characterized by fibroblasts. Combined with the inhibitory effect of α-SMA on myofibroblast migration, a high concentration of TGF-β induced myofibroblast adhesion and high tension, while a low concentration of TGF-β induced myofibroblast migration and collagen secretion. Based on the DMTM model, the initial stage of liver fibrosis includes high tension in the portal vein region and collagen expansion around the portal vein, which can effectively explain the interlobular collagen deposition in the early stage of fibrosis. This represents theoretical progress in the initiation and progression of liver fibrosis. In addition, our discussion further emphasized the important role of targeted TGF-β in the treatment of liver fibrosis.

## Figures and Tables

**Figure 1 ijms-24-15460-f001:**
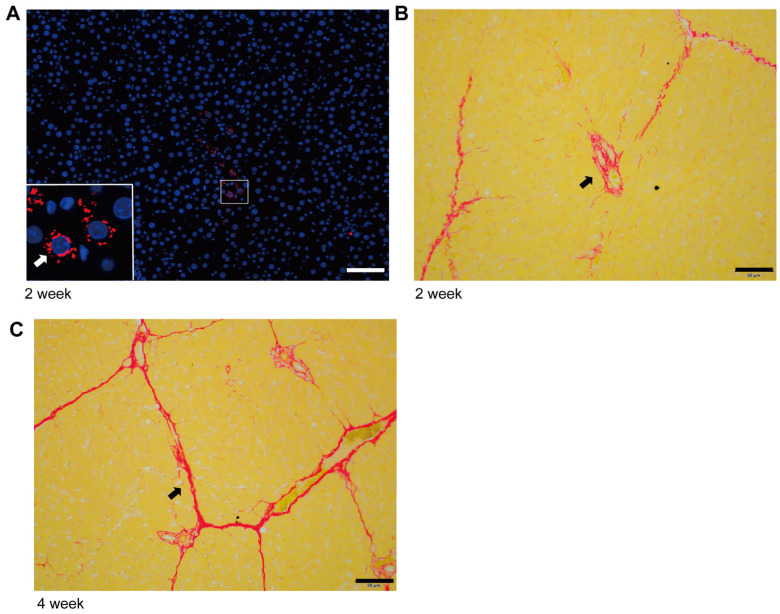
Characteristics of collagen deposition in the initiation and progression stage of liver fibrosis. (**A**) High TGF-β expression (Fluorescence, red) and fibrosis characteristics in portal vein region at the initial stage of liver fibrosis (2 weeks of modeling) (scale bar, 50 µm). (**B**) Characteristics of collagen deposition (Sirius red, red area) in the initial stage of liver fibrosis (2 weeks of modeling) (scale bar, 50 µm). (**C**) Characteristics of collagen deposition (Sirius red, red area) in the advanced stage of liver fibrosis (4 weeks of modeling) (scale bar, 50 µm).

**Figure 2 ijms-24-15460-f002:**
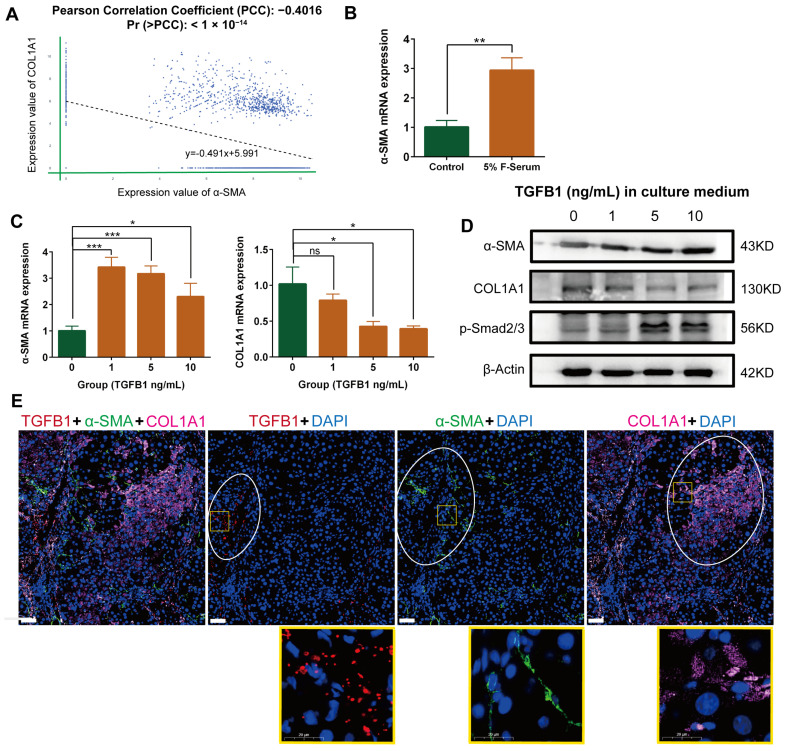
Negative correlation between α-SMA and COL1A1 expression in myofibroblasts derived from hepatic stellate cells. (**A**). Correlation between α-SMA and COL1A1 expression in mesenchyme cells; data from Omnibrowser database. (**B**). The mRNA expression of α-SMA in HSC-T6 cells treated with 5% rat liver fibrosis serum. (**C**) The mRNA expression of α-SMA and COL1A1 in HSC-T6-derived myofibroblasts in the presence of gradient TGF-β1. (**D**) The protein expression of p-Smad2/3, α-SMA, and COL1A1 in HSC-T6-derived myofibroblasts in the presence of gradient TGF-β1. (**E**) The expression of TGF-β1 (red), α-SMA (green), and COL1A1 (pink) in the fibrotic liver of the rat was labeled with fluorescent tricolor (scale bar, 50 µm). The data represent the mean ± S.D. of three independent experiments. *t*-tests were used for statistical analysis. * *p* < 0.05; ** *p* < 0.01; *** *p* < 0.001.

**Figure 3 ijms-24-15460-f003:**
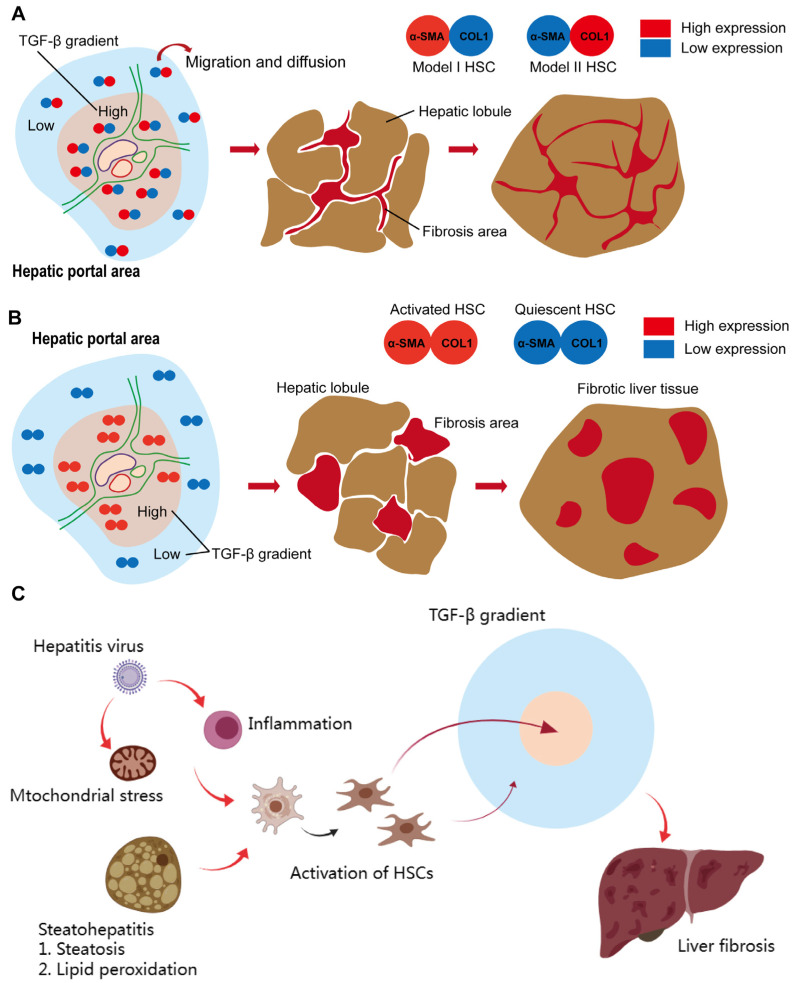
The dual mode transition of myofibroblasts derived from hepatic stellate cells (DMTM model). (**A**) In the DMTM model, myofibroblasts have dual modes. Myofibroblasts obtain high expression of α-SMA and enhanced tension, accompanied by inhibition of COL1A1 expression in the high-concentration region of TGF-β. At the edge of the TGF-β area, myofibroblasts convert to a high-collagen secretion phenotype. The DMTM model emphasizes the relationship between the TGF-β gradient and myofibroblast phenotype, which provides a more reasonable explanation for the formation of network collagen deposition. (**B**) In the traditional HSC activation model, myofibroblasts simultaneously obtain high expression of α-SMA and COL1A1. Based on this model, the concentration gradient of TGF-β1 induces the activation of myofibroblasts in the high-concentration area and their quiescence in the low-concentration area. Activated myofibroblasts secrete collagen components, which leads to collagen deposition and clumping around TGF-β1-positive areas. Based on the traditional model, the tissue characteristics of the fibrotic liver will be the deposition of sheet-like fibers. However, this is inconsistent with the objective situation. (**C**) Model of the initiation and development of viral hepatitis and steatohepatitis-related liver fibrosis.

## Data Availability

The datasets used during the current study are available from the corresponding author on reasonable request.

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
