# Peer review of "The Dual-Mode Transition of Myofibroblasts Derived from Hepatic Stellate Cells in Liver Fibrosis"

_ijms, 2023, doi:10.3390/ijms242015460_

Round 1
Reviewer 1 Report (New Reviewer)
The study was performed to investigate the applicability of novel proposed “dual mode transition model of myofibroblasts in fibrotic liver”(DMTM model) instead of the “myofibroblast with α-SMA high collagen high model”, which appeared to not effectively explain the initial stage and progression characteristics of liver fibrosis. myofibroblasts have dual modes. At the edge of the TGF-β positive region, myofibroblasts were found to convert into a high migration and high collagen secretion phenotype. Authors believe DMTM model provides a reasonable explanation for collagen deposition and expansion in the initial stage of liver fibrosis.
The work is of interest, but there are several questions to be addressed.
Major Comments
1. In the text and in Figures 1 and 2, information on both models (“myofibroblast with α-SMA high collagen high model”and the “dual mode transition model of myofibroblasts in fibrotic liver”(DMTM)) and their differences is not enough. It is highly recommended to make it clear. For instance, to present the detailed explanation of both models in the same Figure, but not in the separate figures. More explanation is also necessary in the text.
2. In the Figure 2E, it is very difficult to see the results with double and triple staining. The magnification is low and the high magnification picture in the corners of the photos is too small. Furthermore, green color alpha-SMA is almost impossible to see. It is better to substitute the figures and to show high magnification clearly.
3. In the Western blot membrane photo with collagen 1A1 there is very high background in the first two bands in the left. Therefore, it is highly recommended to substitute the photo, as it is difficult to see the COL1A1 decrease.
4. There is a discrepancy in the RT-PCR and Western blot results. The highest expression of alpha-SMA mRNA is observed in TGFB1 1 ng/ml group, but the protein expression is the highest in TGFB1 10ng/ml group.
5. In Materials and Methods, in formation an animal experiment is not enough. Animals grouping, treatment, numbers used in the experiment, animals gender, condition, body weight, food and water consumption, must be clearly presented. It is necessary to indicate the age of rats which were subjected to the study. Are there any changes in liver and other organ weights?
6. The explanation of sample collection in the Materials and Methods section needed to be informative. In total, materials and Methods are not informative.
7. The Figure legends must be rewritten to make them more informative and easy to understand.
8. The English of the paper is need to be well-revised by the English-speaking scientist. Please correct the miss types (for instance, line 58: after the comma could not be a word starting from big letter).
English is necessary to make more clear and understandable, as well as correct the miss types.
Author Response
Please see the attachment.

Reviewer 2 Report (New Reviewer)
The main point of this manuscript is the suggestion by the authors that there is a sequence in myofibroblast activation implying that first fibroblasts show high alpha-SMA and low collagen expression followed by high collagen expression and high migration. The authors show convincing data that justify the hypothesis. The methods used appear adequate.None.
Author Response
Please see the attachment.

Reviewer 3 Report (New Reviewer)
The communication "The dual mode transition of myofibroblasts derived from hepatic stellate cells in liver fibrosis" provides an interesting report on the correlation between α-SMA expression and collagen secretion in the development of fibrosis from HSC activation to myofibroblast pohenotype. Results on cells provide promising insights. However several minor drawbacks along the text and missing details about experiments, detailed below, make the paper to require major revision before to be reconsidered for publication.
Minor remarks
The meaning of some sentences slong the text is to be checked and revised and clarified.
Some examples are:
Lines 16-17-“Collagen secretion and high expression of α-SMA related high tension and migration limitation are the main characteristics of myofibroblasts” Please specify “high tension” Also, next sentence: “How these two characteristics define ….”. To comply with “two characteristics” the previous sentence could be better as: “Collagen secretion and high expression of α-SMA with the related high tension and migration limitation are the main characteristics of myofibroblasts”.
Line 18- please specify HSC derived myofibroblasts.
Along the main text – line 42, please specify “q for qHSCs”
Line 90 – please explain “later” in the “…at the early stage of fibrosis, which was not significantly later than that of portal vein fibrosisW” sentence.
Major remarks:
Legends to Figures 1,2, and “3.6. Immunofluorescence” section. Please clearly specify: 1) acquisition conditions at microscope (bright filed for Sirius red, confocal for fluorescence, is it? Also, please provide Leica manufacturer details, as well as for other material suppliers); 2) fluorescence conditions, i.e. colors, for each antigen detected (i.e. α-SMA; TGF-β1; COL1A1); 3) line 277, please specify that DAPI was used for the counterstaining of nuclei.
In Figure 1 images with liver tissue slices stained with Sirius red are shown, to illustrate the advancement of fibrosis. However, no description is provided as to: 1) how and from which animal liver specimens were procured and stained, 2) meaning of Sirius red staining, 3) alternatively, the source of images and their description af from 2).
Discussion- It would be interesting to discuss the results here provided with literature on the two hits hypothesis (Day & James, Gastroenterology 1998; 114: 842-845 DOI: 10.1016/s0016-5085(98)70599-2; and subsequent reports and comments).
Lines 257-259- please remove text
good
Round 2
Reviewer 1 Report (New Reviewer)
Dear Dr. Li and coauthors,
Thank you very much for the careful correction of the manuscript according to my comments.
I think now it could be accepted for publication.
Kind regards,
The manuscript has been edited. Minor changes are recommended.
Author Response
Dear Reviewer,
Thank you very much for reviewing our manuscript entitled, “The dual mode transition of myofibroblasts derived from hepatic stellate cells in liver fibrosis” (Manuscript Number: ijms-2652007) submitted for publication in International Journal of Molecular Sciences. We have received your comments and thank you very much for your approval of the manuscript.
On behalf of all the authors, thank you very much for reviewing our manuscript and giving us this opportunity.
Sincerely,
Corresponding Author Name: Li Xun
Address: Lanzhou University,
No.1 Donggang West Road, Chengguan District, Lanzhou 730030, Gansu, China
Phone number: 86-18393818949
Email: yanmch18@lzu.edu.cn
Reviewer 3 Report (New Reviewer)
The text of the communication "The dual mode transition of myofibroblasts derived from hepatic stellate cells in liver fibrosis" have been revised. However some minor points still need attention, as detailed below, and make the paper to require a minor revision.
- Line 16-17-“ high tension” is still undefined
- Also, please provide details for Leica (i.e. the current name of the producer, town, country)
- Figure 1 Legend is still missing the indication on images from Sirius red stained liver tissue; also, for Figure 1A, please add indication for Fig. 1A(?) fluorescence, Fig. 1A(?) Sirius red.
- As to Sirius red staining procedure, addition of its description at page 8 is appreciated. However, line 261 is wrong and misunderstanding: Sirius red does not stain muscle, since it is always staining in red collagen. It can be used to distinguish collagen septa from yellow muscle fibers (i.e. see: Stain Technol. 1987 Jan;62(1):23-6. doi: 10.3109/10520298709107961 - this indication is obviously for personal information, not to be added to literature).
good
Author Response
Please see the attachment.

This manuscript is a resubmission of an earlier submission. The following is a list of the peer review reports and author responses from that submission.
Round 1
Reviewer 1 Report
The authors highlight the dual mode transition of myofibroblasts with the DMTM model to reveal the collagen deposition and expansion during progression of liver fibrosis. There are some critical issues should be addressed in this review as follows.
1. First of all, the study upon the roles of HSC subpopulations in liver fibrosis has been greatly discussed under the application of single cell sequencing tool. The present manuscript is lack of novelty and just a pale sequel of the previous papers.
2. In Fig. 2, the authors should provide the double-staining data to verify their hypothesis.
3. The authors should also conduct other HF models to confirm their hypothesis since the etiology of HF is heterogenetic.
4. The multiple sources of myofibroblasts have been confirmed. How to distinguish these cells?
The phrases as well as grammar errors in the manuscript require further reorganization and corrections.
Reviewer 2 Report
The authors propose the idea of DMTM model as an alternate to explain collagen deposition and migration phenotype in liver fibrosis. They show that activated fibroblasts with high collagen low alpha SMA are the ones for progression of fibrosis; while initiation is by alpha SMA positive cells. While the idea is interesting, the authors are recommended to address the following queries:
1. Expand on why "α-SMA high collagen high model could not effectively explain the initial stage and progression characteristics of liver fibrosis" in the introduction.
2. The premise of the study needs to be elaborated further in the introduction.
3. Please correct 'kuffer cells' to kupffer cells in line 59.
4. Please add scale bars to the images and provide high magnification images.
5. Please provide promoter occupancy of SMADs on alpha SMA and COL1A1 promoter in fibroblasts upon TGF beta treatments indicated in the manuscript.
6. TGF beta (same concentration in in vitro assays) is known to induce both alpha SMA and collagen. The authors are recommended to show activated TGF beta signaling (pSMAD2/3 staining) that would conclusively demonstrate that the cells in high TGF beta are the ones responding by increasing alpha SMA.
6. Have the authors looked at other collagen isoforms?
Round 2
Reviewer 1 Report
1. The authors need to provide the double staining of alpha-SMA and collagen I to verify their hypothesis.
2. The data of TGF-beta1 and Smad are not enough to represent the distinct groups of cells since fibroblasts could be also activated by the same pathway.
Minor editing of English language required.
Reviewer 2 Report
Authors have addressed most of the comments satisfactorily. Manuscript can be accepted in present form.
Author Response
Dear Editor and Reviewer,
Thank you very much for reviewing our manuscript entitled, “The dual mode transition of myofibroblasts derived from hepatic stellate cells in liver fibrosis” (Manuscript Number: ijms-2570461) submitted for publication in International Journal of Molecular Sciences. On behalf of all the authors, thank you very much for reviewing our manuscript and giving us this opportunity.
Sincerely,
Corresponding Author Name: Li Xun
Address: Lanzhou University,
No.1 Donggang West Road, Chengguan District, Lanzhou 730030, Gansu, China
Phone number: 86-18393818949
Email: yanmch18@lzu.edu.cn
Round 3
Reviewer 1 Report
Base on the results of double staining, high concentration of TGF-beta seems to cause cytotoxicity and therefore the expression levels of both α-SMA as well as collagen were significantly diminished. This finding is not consistent with the Western blot and their hypothesis.
Minor editing of English language required.